# Gender-Responsive Design of Bacteriophage Products to Enhance Adoption by Chicken Keepers in Kenya

**DOI:** 10.3390/v15030746

**Published:** 2023-03-14

**Authors:** Zoë A. Campbell, Nelly Njiru, Amos Lucky Mhone, Angela Makumi, Sylvain Moineau, Nicholas Svitek

**Affiliations:** 1International Livestock Research Institute (ILRI), P.O. Box 30709, Nairobi 00100, Kenya; n.njiru@cgiar.org (N.N.);; 2Département de Biochimie, de Microbiologie, et de Bio-Informatique, Faculté des Sciences et de Génie, Université Laval, Québec, QC G1V 0A6, Canada; 3Félix d’Hérelle Reference Center for Bacterial Viruses, Université Laval, Québec, QC G1V 0A6, Canada

**Keywords:** gender-responsive, phages, poultry, veterinary products, antimicrobial resistance (AMR), zoonoses, vaccines, bacterial disease

## Abstract

Women and men keeping chickens in Kenya aspire to have a source of income, feed their families healthy food, and grow their businesses. Managing animal diseases and minimizing input costs enable their success. This study uses qualitative methods to recommend design opportunities for a veterinary product under development in Kenya that contains bacteriophages (phages) that target pathogenic *Salmonella* strains responsible for fowl typhoid, salmonellosis, and pullorum in chickens and foodborne illness in people. Our findings revealed the interplay between gender and two production systems: free-range and semi-intensive. Chicken keepers in both systems could benefit from phages combined with the orally administered Newcastle disease vaccine, one of the most commonly used preventive veterinary interventions, or phages as a treatment for fowl typhoid. Oral administration is less labor intensive, with greater benefits for women who have less control over family labor and reported doing more care tasks themselves. Men in free-range systems usually pay for veterinary inputs. In semi-intensive production systems, a phage-based product used prophylactically could be an alternative to expensive, intramuscular fowl typhoid vaccines. Keeping layers was common for women in semi-intensive systems, as they are more economically impacted by reduced laying caused by bacterial diseases. Awareness of zoonoses was low, but men and women were concerned about the negative health effects of drug residues in meat and eggs. Therefore, highlighting the lack of a withdrawal period for a phage product may appeal to customers. Antibiotics are used to both treat and prevent diseases, and phage products will need to do both to compete in the Kenyan market. These findings guide the ongoing design of a phage-based product with the goal of introducing a new veterinary product that meets the diverse needs of chicken keepers in Africa and serves as an alternative or complement to antibiotics.

## 1. Introduction

Globally, the poultry industry has increased 5% every year for the past three decades [1]. One report suggests that egg production in developing countries increased by 331% from 1980 to 2000 [2]. In Kenya, poultry farming represents about 30% of the total agricultural contribution to the Gross Domestic Product, with an estimated 75% of rural families keeping chickens, with an average of 13 birds per household [3]. Chickens are important assets to both women and men; their roles include improving financial and food security and strengthening social relations, such as when given as gifts or slaughtered for visitors [4]. Women dominate small-scale chicken production [5], and they often control the sale of chickens and the proceeds [6]. These proceeds are a unique source of livelihood for women whose caretaking responsibilities can limit their income-generating opportunities [7]. This income is threatened by infectious diseases that reduce productivity. Infectious diseases associated with chicken farming and egg production also pose a risk to the health of livestock keepers and consumers. Women and youth are at a higher risk of exposure to infectious diseases than men because of their direct involvement in routine care [8]. Nontyphoidal *Salmonella* is a major cause of foodborne infections in both developed and developing countries [9]. In low-income countries, it can also evolve into a life-threatening and invasive form of the disease in humans, which occurs in young children at rates of up to 388 cases per 100,000 people [10]. Africa has a high number of foodborne diseases, with approximately 91 million related diseases and 137,000 deaths per year [11]. Some studies on *Salmonella enterica* serotype Typhimurium, the main cause of salmonellosis in humans, suggest poultry are carriers [12], while other studies question the findings, showing that some serotype variants have a narrow host range and cause disease only in humans and higher primates [13]. In chickens, *Salmonella* causes fowl typhoid, salmonellosis, and pullorum disease [14,15]. Fowl typhoid is an acute septicemic condition that mainly affects mature birds, especially commercial layers. Salmonellosis, caused by the same pathogen that causes gastrointestinal infections in humans, can cause enteric disease in chickens, with signs such as diarrhea. It is often self-limiting in chickens, with the possibility of infection without showing clinical illness. Pullorum disease causes diarrhea and mortality in chicks and decreased egg production in adult birds. 

Current methods for controlling *Salmonella* infections in poultry farms include the use of antibiotics to treat and prevent infections. Antibiotics may also be used at subtherapeutic doses to increase the growth rate of chickens and improve feed efficiency. However, a study of chicken farmers keeping layers in Ghana and Kenya as well as broilers in Zambia and Zimbabwe found limited use of antibiotics for growth promotion, with treatment and prevention as the most common reasons for use [16]. 

Antimicrobial resistance (AMR) is a growing concern in *Salmonella* infections [17]. An estimated 75% of antibiotics administered to poultry are released into the environment and likely contribute to the emergence and spread of AMR [1]. While it is difficult to estimate the prevalence of antimicrobial resistance in Sub-Saharan Africa due to limited surveillance programs and research, studies such as those by Langata et al. (2019) have found evidence of antibiotic-resistant *Salmonella* isolates. Isolates from chicken droppings in Nairobi, Kenya, showed resistance to amoxicillin, cotrimoxazole, tetracycline, and streptomycin (50%, 28%, 11%, and 6% of the isolates, respectively [18]. None of the isolates were resistant to gentamicin, nalidixic acid, ciprofloxacin, or chloramphenicol. Another study reported that *Salmonella* isolates from poultry feed in Kenya had the highest resistance to amoxicillin (41%) [19].

Among strategies to control bacterial infections, the use of bacterial viruses, or bacteriophages (phages), is on the rise as an alternative or complement to antibiotics [20,21]. Phages are more specific than antibiotics, often infecting and killing a single bacterial strain or a group of related strains, limiting their impact on the microbiota. They also have the advantage of co-evolving with their bacterial host, reducing the emergence of long-term resistance [21]. In recent years, several phages targeting *Salmonella* have been isolated, and some have been used to treat *Salmonella* infection in chickens [21,22,23,24,25,26]. There are already at least seven different commercially available *Salmonella* phage-based products that have been registered in the USA, Poland, China, Colombia, Chile, and in Southeast Asian countries [27]. A number of them showed efficacy on food products and poultry meat [28]. At this time, a commercial product has yet to be made available or registered in Kenya. Several institutes, including the International Livestock Research Institute (ILRI), are working on developing phage products and cocktails to tackle AMR in Kenya, as demonstrated by the recent Kenyan Phage Symposium held in 2022 [29]. There is openness to licensing or registering a phage-based product in Kenya; Makumi et al. (2021) describe initial discussions with the Veterinary Medicines Board [30]. In chicken production, phages could not only be used for treatment and prevention but also as a surface cleaner that targets specific bacteria. A recent review discusses the potential benefits and challenges of using phage therapy within the Sub-Saharan African market, including the creation of phage biobanks and the regulation of phage products for veterinary use [30]. Considerations include how a product could be registered; for example, phages alone would not be considered a vaccine even if used prophylactically because they do not induce an immune response. A document outlining guidelines for veterinary phage product registration by the European Medicines Agency acknowledges that “due to the specific nature of bacteriophage products, adaptation of the general rules may be acceptable, and the regulatory framework is expected to be flexible” in part because of the multiple potential applications and the potential to co-administer phage products with other veterinary products such as antibiotics [31]. 

To develop a product for the Sub-Saharan market, it is important to understand the conditions under which poultry production occurs. In low- and middle-income countries such as Kenya, poultry production, even in urban areas, is dispersed across many farms and households, many using free-range and semi-intensive production systems. In Sub-Saharan Africa, estimates of the total poultry population kept in rural or traditional production systems range between 60% and 80% [2,32]. Chickens in these production systems are often under the control of women and are an important economic and nutritional resource for women and children [2]. In Kenya, indigenous chickens, often associated with more extensive production systems, made up half of the value of the national chicken meat production in 2004, with the other half coming from broilers and culled layers [3]. In this context, chicken production is both a business and a household enterprise where household members, mostly women and youth, contribute labor and business expenses and profits are intertwined with the financial needs of the household. Veterinary decision-making happens within a social context influenced by gender norms and intra-personal relationships between household members and veterinary providers. Women play an active role in poultry management, particularly in Sectors 3 and 4, as defined by the Food and Agriculture Organization of the United Nations (FAO) [3], which we refer to as semi-intensive and free-range production systems, respectively. Even in Sector 3, which has semi-intensive production systems, biosecurity is low; chickens may spend time outside in contact with wild birds or other livestock, and the chicken owners may still be vulnerable to food insecurity.

Management practices, the division of labor, and access to veterinary services vary by gender. Women and men have different levels of decision-making ability and access to resources to prevent or treat bacterial infections and other animal health issues [8]. In a study about Tanzanian smallholder chicken-owning households, there was no difference in awareness of Newcastle disease vaccines or their use between households with men or women as decision-makers for chickens [33]. The same study, however, reported more use of aloe vera to prevent or minimize signs of Newcastle disease in households where a woman was the primary decision-maker for chickens (41% vs. 29% in households where a man was the primary decision-maker). Women’s greater reliance on aloe vera may suggest less access to veterinary products, either because of their cost or because of differences in knowledge and access to veterinary services and information. A study on the use of insects as feed for poultry and fish in Kenya showed that while men and women were both able to list insects as appropriate for feeding chickens, women knew of more commercial and homemade feed types compared to men but were less able to purchase commercially available feed because of its cost [6]. Levels of access to information and service provision have also been shown to vary between genders and affect the practices used. In a study on antimicrobial use in poultry systems in four African countries, women had a 5% lower knowledge score (*p* < 0.05) on a set of questionnaires about best practices in antibiotic use than men when other variables were held at their mean and controlling for country [16]. A survey conducted by the FAO, covering 97 countries with sex-disaggregated data, looked at gender equity in extension, referring to services delivered by experts on agriculture, agribusiness, and health. Only 5% of all extension resources were directed at women. Moreover, only 15 % of the extension personnel were women [34].

Given the documented gendered differences in the division of labor and the access to information and veterinary services in the poultry sector, it has been highlighted that a gender lens is relevant when considering the design of a new veterinary product. This study uses the framework for gender-responsive animal health research developed by ILRI, which recommends the consideration of gender at three stages of research: research prioritization, product design, and delivery of products and services [35]. Using this framework, we ask the following question: What design considerations of a phage-based product can potentially improve adoptability by female and male chicken keepers with smaller flock sizes in semi-intensive and free-range production systems? The objectives of this study were to characterize chicken production by gender in two production systems (free-range and semi-intensive); map gendered disease problems and existing solutions; and identify product design opportunities with the goal of improving uptake of a phage-based product or products. A parallel activity within the same project is the biological characterization of known or novel phages towards Kenyan *Salmonella* strains, which is underway and will be presented in upcoming manuscripts.

## 2. Materials and Methods

### 2.1. Study Sites

Six sub-counties in Nairobi, Kiambu, and Machakos counties in Kenya were purposefully chosen to overlap with the sites where phages were isolated for product development in the larger research project. The locations are also ideal because proximity to Nairobi increases chances of accessing a new veterinary product, but a focus on semi-intensive and free-range production systems highlights the needs of chicken keepers, who may be less likely to benefit than more industrial and intensified chicken operations. Nairobi County contains the capital city of Nairobi. With a high population density of 4515 people per square km in 2009 [36], the area hosts private businesses catering to livestock keepers, from feed companies to agro-veterinary supply shops (agrovets), and many urban consumers of eggs and meat. Kiambu County is directly adjacent to the city of Nairobi, with a large urban population because of the northern growth of Nairobi and a population density of 638 people/square km. The Kikuyu people are the dominant ethnic group. Machakos County shares a western border with Nairobi and Kiambu counties, with the county capital, Machakos, located about 65 km away from Nairobi. Machakos County has a larger rural population and a lower population density (177 people/square km) compared to Nairobi and Kiambu Counties. The Kamba are the dominant tribe. The poverty rates are much higher in Machakos County, with 57% of the population at or below the national poverty line, compared to 22% and 25% in Nairobi and Kiambu Counties, respectively, as per 2006 data [36].

Veterinary services in Kenya are provided publicly and privately. In the early 2000s, government extension services in Kenya, including veterinary services, shifted to a demand-driven model [37]. Consistent with the national model, participants within the study counties reported accessing veterinary services from public and private providers.

### 2.2. Data Collection

A participatory, qualitative methodology was employed for this study, highlighting the needs and current practices of chicken keepers within their social and economic context. This approach acknowledges the complexity of the decision about whether to adopt a new veterinary product or not. Data collection methods consisted of 24 focus group discussions (FGDs), with chicken keepers disaggregated by gender (women and men) and production system type (free-range and semi-intensive), and 17 key informant interviews (KIIs) with experts who have professional-level knowledge of the poultry sector and work closely with chicken keepers (see the demographic breakdown in the results section). Field activities were conducted in April 2021. The FGD and KII tools are available in the Field Researcher Guide [38].

The sample size considered the goal of gathering the following: perspectives from women and men from three regions that are users of two production systems; the number of people required for an engaging and inclusive discussion; and the likelihood of reaching data saturation for the topics of interest. Eligible participants were women and men (at least 18 years old) who owned, made decisions about, or cared for chickens on their farm or household, had chickens within the past 3 months, and fit the respective production system criteria. Free-range production was defined as fewer than 50 chickens; semi-intensive production was defined as 50 or more chickens. Participants were chosen purposefully with help from community leaders and local government officials, with recruitment for each production system based on the approximate flock size the participants “typically” have, in acknowledgement of the cycles of buying and selling within chicken production. As a result, 12% of participants reported a flock size outside the desired threshold specified for their FGD at the time of data collection (ten outliers for the free-range FGDs and nine outliers for the semi-intensive FGDs). Unless the participants failed to meet other eligibility criteria, they were retained in the study because removing them after recruitment could have incentivized lying about flock size or prompted arguments about their “typical” flock size after recruitment by the community mobilizer and travel to the meeting location. The relatively low threshold for flock size targeted chicken keepers corresponding to FAO’s definition of Sector 4 (village or backyard) and Sector 3 (commercial) chicken keepers [3]. The group size was capped at eight participants due to government-imposed restrictions on large group meetings during the COVID-19 pandemic. 

All research activities were facilitated in Swahili or English, depending on the participants’ preferences, with a man facilitating the men’s FGDs and a woman facilitating the women’s FGDs. Typically, two people took independent notes, but occasionally a single note-taker took notes for key informant interviews. Participants were asked to give written consent to participate. Audio recordings were taken for all activities. At the end of the activity, participants were reimbursed a modest amount for their travel to the meeting point and provided with drinks and snacks to consume at home as a COVID-related alternative to refreshments typically provided during the discussions.

### 2.3. Lean Social Canvas

The discussion tools for the FGDs and KIIs were based on the social lean canvas template, a tool used to organize early-stage ideas for a social enterprise by identifying problems, existing solutions, and the unique value propositions of new products [39]. Guided by the lean social canvas template, we asked what problems each group currently faces in chicken keeping. These problems may or may not be directly related to bacterial diseases. Next, we considered whether phages can provide a solution to any of the existing problems, identified the existing alternatives that the respective group of chicken keepers is currently using to address the problem, and suggested a unique value proposition, or “selling point”, that a phage product can provide for each group.

While the discussion tools were designed to assess the feasibility of a phage product, we did not mention phages directly or ask for the participants’ opinions about phages. An informed end user of a veterinary or medical product can reasonably expect information about the product’s efficacy, safety (including side effects), and usage, but further details may be beyond the scope of what is useful. Since previous knowledge of phages was unlikely, any discussions would have involved the research team introducing the concept for the first time, and attempting to gauge opinions shortly after would likely be both unreliable and distracting. The tool focused instead on broader considerations affecting the feasibility of a new veterinary product or products. Phages were mentioned by name with key informants, whose technical backgrounds allowed them to appreciate the additional detail.

Relevant context included reasons for keeping chickens, breed preferences, roles in routine and management activities, how tasks are divided amongst family members or employees, animal health and production challenges, and awareness of zoonotic diseases. Within this context, we narrowed in on the use of veterinary products, discussing gendered access to veterinary services and inputs, relationships with animal health professionals, currently used veterinary products, and preferences for new products. The number of FGDs where a topic is mentioned out of the total number of FGDs for that specific group is included in parentheses. An FGD was counted if at least one participant in the group raised the topic. The KIIs triangulated the information collected from FGDs.

### 2.4. Analysis

The recordings were fully transcribed and translated into English. Complete transcripts of all FGDs are available online [40] in support of the movement towards more open data in qualitative research [41]. Key informant interview transcripts are not shared to protect the anonymity of the interviewees. A final document, containing the transcriptions plus relevant notes (such as body language or details about the group dynamics), was created for each research activity and coded thematically using NVivo (released in March 2020) [42]. Most of the coding was deductive, using a predetermined coding framework, with some opportunity for inductive coding through adding themes that emerged from the data in a second iteration of the coding framework. Disease and health concerns were coded by the disease name for commonly recognized diseases, such as the Newcastle disease and respiratory illnesses, and by signs relevant to bacterial diseases, such as white diarrhea. Two people worked together to code the documents. Each document was coded by only one person; however, both coders trained on a subset of documents and discussed discrepancies to ensure the themes were understood and being coded similarly. Since identical FGD questions were used, analyses compared the content of information on themes and the numbers of mentions across themes to identify differences and similarities by gender and production system. The final coding framework is available in Appendix A, Table A1. Quotations are identified with the gender of the participant, their production system, and county either within the text or in parentheses directly after the quote.

## 3. Results

The results are organized as follows: we start by providing a characterization of the study participant and the two production systems (free-range and semi-intensive); we then provide an overview of the gendered management of chickens; animal disease concerns and solutions; access to animal health services; use of animal health products (including vaccines, drugs, and traditional medicine); and opportunities and barriers to accessing animal health products.

### 3.1. Participant Demographics

We conducted 24 focus group discussions with 162 chicken keepers (85 women and 77 men) (Table 1) and 17 interviews with key informants. Informants included public and private veterinarians, government-employed livestock production officers, a hatchery employee, and employees and owners of agrovets, shops that sell farm inputs and veterinary products. The gender distribution among the 162 chicken keepers participating in the focus group discussions across the three counties and six sub-counties was relatively equal, with 53% women and 47% men. Most participants were married (86%) with secondary school education or above (83%), and between 35 and 60 years old (67%), as shown in Table 2. The youth participants (<35 years old) were disproportionately male (66%). The key informants included seven men and eleven women (one interview had two informants).

### 3.2. Reasons for Keeping Chickens and Breed Preferences

The decision to adopt a new veterinary product can be better understood in the context of people’s financial and personal incentives for keeping chickens. In this section, we discuss motivations for keeping chickens, preferences for types of chickens (broilers, layers, improved local chickens, and local chickens), and preferences for specific breeds, all of which can influence the need for veterinary input and perceptions about vulnerability to disease.

There are common reasons for women with free-range production systems (<50 chickens) to keep chickens for eggs and meat and sell them to cover modest financial needs in the household. As expressed by a woman in Machakos County:
*I keep chickens because of their many benefits. I take them to the market when I do not have money, sell them, and use the money to do things at home. They give me eggs that my children eat, and at times I sell the eggs. They make my home beautiful, and they give me manure.*

For men with free-range production systems, income and making money were more emphasized as reasons to keep chickens, although household consumption of meat and eggs was also mentioned. As a man from Nairobi County explained, “I keep chickens because, most of the time, when they mature, I make money from them.” 

Women with semi-intensive production (>50 chickens) referred to chicken-keeping as their business, also citing personal interest or passion as a motivator. A middle-aged woman in Kiambu County, a university graduate, explained. “I like farming so much. I like trying different things and exploring. At first, I tried broilers and pigs. I was doing it because I liked agriculture, but now I do it as a business.”

Men typically keeping more than 50 chickens described keeping chickens for income. A Kiambu County participant referred to chicken-keeping as his “side hustle.” A farmer with secondary school education in Kiambu County elaborated. “The reason for raising chickens is to sustain myself in life and build commercially by taking on a project.” 

Even as levels of intensification increase, local breed chickens (referred to as *kienyeji* chickens in Swahili) and improved local breeds (breeds with genetic contributions from both exotic and local chickens) feature prominently because of their reputation as disease-resistant compared to “exotics” such as layers and broilers. In addition, many consumers in Kenya prefer meat and eggs from local and improved local chickens. Improved local breeds mentioned by name in the FGDs included Sasso (three FGDs), Kenbro (three FGDs), and Kuroiler (two FGDs). Kari and Rainbow roosters were mentioned only by key informants.

Table 3 shows the types of chickens kept by the 162 participants. Many of the semi-intensive producers kept multiple types of chickens. For men, the most common combination was local and improved, kept by 23% of all participants (9/40), and for women, layers and local chickens were most often combined, reported by 15% of participants. The median flock size for women with semi-intensive production systems was higher (250 chickens) than that of semi-intensive men (only 70 chickens).

As a woman with a free-range production system in Nairobi County explained: “I stick to *kienyeji* because the meat is tastier. Their eggs are also tastier, with a yellow yolk.” A man, also in Nairobi County, described the market demand: “Me too; I keep *kienyeji* like my colleagues because feed prices have gone up and these *kienyeji* chickens can be fed maize from the market. I can chop greens for them. They feed well according to one’s financial ability. They are very marketable, especially during Christmas time, when they are in high demand.” A woman in Kiambu County with semi-intensive production described her shift from keeping broilers to keeping layers:
*I kept broilers for almost ten years. They are marketable. When you are consistent, you can keep them because the people you supply are asking for the chickens, but we were not consistent. Sometimes we did not have them, so I turned to layers, and it is going very well. The only challenge is that before they start laying the eggs, they spend a lot of money, but otherwise the layers are better.*

For men with semi-intensive production, concerns about disease were a motivating factor to prefer local or improved chickens, with participants in all six FGDs mentioning they are perceived as “resistant,” “less prone to disease,” and “their sickness is not comparable to broilers.” Local and improved chickens were also perceived by both women and men to be more responsive than exotic chickens to readily available and inexpensive traditional medicines like aloe vera. 

Local chickens were also associated by the participants with healthier meat and eggs, in part because of concerns about the human health effects of drugs and other veterinary inputs used in rearing exotic chickens. This came out most strongly in the men’s free-range FGDs in Nairobi County. Thus, in addition to taste preferences, health considerations likely explain the enthusiasm for local chicken products, especially among urban customers.


*On the subject of side effects, I personally do not like keeping exotic breeds because of the additives in feeds used to stimulate growth. There are effects from these exotic breeds; for instance, a three-month-old chicken is ready for consumption. Even us in the village may be at risk because we will find them all over the marketplaces, so I fear them so much.*



*I want to say that kienyeji chickens have no effect. I have been with them for about 40 years; they do not need a lot of medication and hence are safe to eat.*


### 3.3. Roles in Routine and Management Tasks

Understanding who conducts routine and management tasks within the household is relevant for understanding who to target with a new veterinary product. Routine tasks include cleaning, feeding, and daily chores, while management tasks include buying and selling chickens as well as veterinary decision-making such as purchasing and administering veterinary inputs.

#### 3.3.1. Free-Range Production

Women and men with free-range production systems described doing routine tasks themselves or with the help of other household members, usually school-aged boys and girls or women employed as house help. Men who spent less time at home delegated tasks to their wives or children, although some older men reported doing routine tasks themselves. Two common areas of co-investment by husbands or other adult family members were purchasing feed and building/expanding chicken housing, discussed further in the production factors section. When it came to buying and selling, responsibility in most homes narrowed to husbands and wives, with women in Machakos County describing more consultation with and permission-seeking of their husbands. In a few cases, other adult household members, such as a daughter, also participated in decision-making. A Nairobi County man in his sixties served as a reminder that decision-making authority and ownership do not always perfectly overlap. “I say they are mine, but even if I say that, my wife is at liberty to sell or eat them. She asks whether she can sell/eat, and I cannot refuse because the household belongs to both of us.” 

In a Machakos County FGD, two participants described the gender norms in their area, the first saying: “Many men here do not keep chickens. Women are the ones who keep chickens.” This prompted the second woman to explain: “Or maybe the wife died, or he has children, and they are taking care of them. However, most of the families are complete families, so the chickens are owned by the women.” This observation is reflected in the participant demographics: 86% of all participants were married (Table 1).

When asked about veterinary decision-making, most discussions in the free-range production FGDs cantered around treating chickens rather than seeking preventive care. Women usually identified the sick chickens and administered the drugs, an in many cases husbands or other family members paid for the administered drugs. A self-described housewife in Machakos County with about 60 chickens explained: “My husband is not near, but when I get a sick chicken, I tell him, and he brings the medicine. Before he comes, I take aloe vera and give it to them.” A woman in Nairobi County reported traveling to the agrovet and purchasing herself because she felt a sense of urgency.


*When I wake up in the morning and find my chickens sick, I am not happy, and I do not have peace. I will act fast to get the medicine to give to them. If I send my husband, he may have other things to do, and he may take time to bring it, making me wait to give the chickens medicine. Sometimes he may even forget.*


When speaking about veterinary decision-making and sourcing veterinary inputs, men described cooperating with their wives to organize seeking treatment. They made more references than the women with free-range production to “knowing vets and agrovets,” “having a guy” who delivers and administers drugs, and “having the phone number of one of the vets.” Almost all of the men described paying for any veterinary inputs themselves, although a few said their wives, or in one case, their mothers, might look for inputs if they were more available.

#### 3.3.2. Semi-Intensive Production

Women and men with semi-intensive production systems (>50 chickens) described a greater variety of routine tasks, including washing drinkers and feeders, collecting eggs, and replacing sawdust. Women, who in this study had larger flock sizes than men, mentioned more tasks associated with larger flock sizes, such as checking incubators, spraying disinfectant, and slaughtering chickens for market.

Women described doing routine activities with assistance from family members, including a brother, mother, and school-aged children, and in some cases, with employees. A single woman in Nairobi County described hiring people “for some hours or as we have agreed”, while a woman in Machakos County had a male employee who did routine tasks most days. A businesswoman in Nairobi County explained that profit margins do not allow her to hire an employee. “I do it because the profit margin for broilers is very low. If I employed someone with a monthly salary, I would get very little profit.”

While some men performed routine activities, they reported a greater reliance on their wives, family members, and household employees. A teacher in Kiambu County with 1200 layers described how he schedules his day with an employee: “I decided to feed them once a day in the evening because I am working, and you realize at some point you may not have an employee. If that happens, will I still manage to feed them after work?” When it came to identifying sick chickens, anyone noticing a problem was expected to report it, but most men felt it was their responsibility to check the chickens frequently, confirm if a problem had been reported, and liaise with a vet or agrovet.

Women with semi-intensive production reported having significantly more authority to buy and sell chickens without consulting their husbands or other family members. Furthermore, a woman in Kiambu County told us that her husband supports her when customers purchase chickens from her early in the morning:
*There’s consultation here and there, but I solely make the decisions. I do inform him that I made this decision because of this and that. It is a formality because he is not interested. Sometimes the chickens are being picked up at 4 a.m., and the ones taking them are men. I have to tell him to be available that day because I am selling chickens and they will be picked up at night. Even if he will not support, he has to be there for security. Consultation comes in handy.*

In contrast to the free-range production FGDs, more of the discussions about veterinary decision-making focused on vaccination than on treatment. Women reported sourcing vaccines and medications from agrovets, paying for them, and administering them themselves, except in the case of intramuscular injections, when they engaged veterinarians. A few worked with employees, describing teaching them about the administration of vaccines, drugs, and vitamins, but always under close supervision. “Apart from collecting eggs, for things like vaccinating, I ensure that I do it, although when I am doing it, she (an employee) is there to learn”, said a woman in Kiambu County with 900 layers and improved chickens. In a few cases, women reported their husbands paying for veterinary inputs. A man in Kiambu detailed the financial understanding he has with his wife when asked who purchases drugs and vaccines: “Mostly it is my wife. We have structured it so I cater for items like feeds that need huge finance, but my wife does the ones that are less capital intensive.” 

For many of the chicken-keeping enterprises, the financial structure is still linked to the financial structure of the typical African household, whereby the man as head of household is expected to provide resources and take on the risk of economic activities conducted by other household members. A man in Kiambu County pondered the complexity of ownership and responsibility:
*There is a saying from my past that ‘the bedbugs and the lice in a house belong to the man of the house’. Hence, the chickens belong to the man of the house, and he can delegate responsibilities to anyone. My wife is the owner, unless she gets stuck and requests for me to boost her. Sometimes there are losses, and I cannot say that my wife should bear the loss alone. We sit down and agree. Since it is a family thing, we decide as a family.*

### 3.4. Disease and Health Concerns

The most frequently mentioned health concerns were the Newcastle disease (ND), raised in 16 of the 24 FGDs, followed by respiratory illness (15 FGDs), coccidiosis or bloody diarrhea (13 FGDs), parasites (13 FGDs), fowl typhoid (6 FGDs), and salmonellosis (3 FGDs), as shown in Table 4. Mentions of coccidiosis and parasites were highest among men in FGDs in the semi-intensive system. Pullorum disease was not mentioned in any FGDs. Since chicken keepers could not reasonably be expected to make veterinary diagnoses or know fewer common diseases by their scientific names, we coded mentions of three signs of disease associated with but not exclusive to bacterial diseases: white diarrhea (11/16 FGDs), chick mortality (7 FGDs), and reduced laying (4 FGDs). Most health concerns were present across all FGD categories, with the exception of white diarrhea, which was mentioned mostly by women with free-range production (five FGDs) and least by men with semi-intensive production (one FGD). 

A government-employed livestock production officer in Machakos County summarized common diseases for the chicken keepers he works with: “Newcastle disease is the starting point, with fowl pox and Gumboro (infectious bursal disease) also coming up. Fowl typhoid is another one.” A livestock production officer in Kiambu County agreed she receives fewer complaints about bacterial diseases but shared an interaction with one farmer who suspected salmonellosis.


*I have not had so many complaints about that (bacterial diseases), because mostly it is a hygiene issue. Even when they die of that, they (farmers) will not know. There is one farmer who called me. He was sure it was salmonellosis because he is young and he may be able to Google. You know, Google has become the best vet and the best doctor. For many, it is Newcastle and Gumboro.*


When chicken keepers spoke about bacterial diseases, fowl typhoid (caused by *Salmonella* gallinarum) was the most commonly mentioned by name. Participants with free-range production systems described clinical signs and difficulty treating them, while commercial keepers mentioned the vaccine. A man in Kiambu County explained he associates fowl typhoid with drug resistance because of previous negative experiences. 


*There was a time I lost a batch of 500 layers. What I have realized is that when fowl typhoid sets in, it is very resistant to drugs. It is like it is suppressed for some time, but even before the dose is finished, it recurs. Twice I have experienced the same, and it is one disease that I have come to fear.*


A semi-intensive producer in Nairobi County was familiar with fowl typhoid vaccines and described access challenges: “You might be using them for the treatment, but they do not work. The drugs are expensive and come in large quantities. The same vaccines are not stored properly.” Another barrier to vaccinating against fowl typhoid is the intramuscular administration, as explained by a woman keeping chickens commercially in Kiambu County: “When I have chicks, the important thing is the vaccine schedule we get from the agrovet, and we give it to them, but for the injections, we call the veterinarian to inject for fowl typhoid and fowl pox.” 

For the few who mentioned bacterial diseases by name, they usually associated them with diarrhea and described purchasing “drugs” or “medicine” to treat their chickens, often after consultation with an agrovet or vet. A man in Kiambu County who attended the free-range production FGD but had previously kept layers on a larger scale explained: “At times, you could find them having diarrhea. When I get this, I consult my vet, and I get advice on what to do. Mostly, he insists on cleanliness.” A woman in Kiambu County with 35 local chickens remarked: “They get diseases fast. When they start to have diarrhea, I buy and give them drugs.”

### 3.5. Relationship with Animal Health Professionals

The participants from all locations described relied on people offering animal health related services for advice and veterinary input. The most common were agrovets, who sell veterinary inputs and give technical advice, and veterinarians, particularly government officers. In Kenya, veterinary officers focus on disease surveillance, vaccination, sample collection, and outbreak reporting, while livestock production officers focus on extension and animal husbandry trainings. Agrovet shops must be licensed to operate, and while there are minimum educational requirements for the owner, employees have varying levels of training and experience. The most common source of veterinary care for chickens was agrovets for women and men in free-range production (55% and 51%, respectively) and commercial men (50%), with veterinarians as the second most common source (Table 5). A young, single man in Machakos County who keeps his own chickens and maintains a flock of chickens for his employer described his strategy if any of the chickens appear unwell: “I am the one in charge of the chickens. I consult with the agrovet and get advice and treatment.” Women with semi-intensive production, the group of participants with the highest flock sizes in the study, reported veterinarians as the most common source of veterinary care (56%), followed by agrovets (41%). 

Vets were described as needed for the administration of injectable vaccines or drugs, especially by commercial women and men with larger flock sizes. A woman in Thika with 400 layers said, “I give them the ones put in water, but there are others that need to be performed by the veterinary officer, especially those for injecting on the wing or thigh.” In Kenya, fowl pox vaccine is typically administered through a wing web stab and fowl typhoid vaccine through an intramuscular injection, usually on the thigh [43]. 

No relationship is perfect, and chicken keepers, veterinarians, and agrovets reported sources of tension. People keeping chickens said animal health professionals were not always available when their services were needed and questioned the technical expertise of some agrovet employees, whom they perceived as lacking in expertise in poultry disease. A few commercial men in Nairobi complained veterinarians had introduced diseases onto their farms by disregarding basic biosecurity practices after visiting other clients. 


*When there is a health issue, I look for people trained in poultry matters, although they are very few. For example, in the village you might find only two of them, and it is very difficult to get them. (Man, semi-intensive production, Machakos County)*



*The veterinary officers from the government know better, and we prefer them because they know what to give when the chicken is sick instead of going to the agrovet, where they do not know what the drug is for. (Woman, free-range production, Kiambu County)*


Government officers expressed frustration that some chicken keepers, especially those with smaller flocks, ask for advice without heeding it and seek treatment advice at the last moment rather than proactively preventing problems through vaccination and good husbandry. They wished more Kenyans understood that veterinary services are now demand-driven and agreed with the chicken keepers’ assessment that not all agrovets are qualified. Agrovets said some chicken keepers arrive to the shop demanding a product recommended by a friend or previously used without understanding the root of the health problem.

Consequently, while agrovets and veterinarians were the primary sources of veterinary care, sources of information about chicken keeping were varied, including fellow farmers, neighbors, friends, and community members; trainings (conducted by county government, feed companies, or NGOs); television programs; internet searches; social media groups; farmer groups; newspapers; radio; feed company extensionists; and expositions such as those hosted by the Agricultural Society of Kenya. 

### 3.6. Veterinary Products

The introduction of a new veterinary product requires understanding existing problems and the products that chicken keepers are currently using. From most to least mentioned, FGD participants mentioned vaccines (22/24 FGDs), traditional medicine (21 FGDs), antibiotics (17 FGDs), and vitamins (10 FGDs). All products were mentioned in all four FGD categories. We also considered cleaning products because one potential application of phages is in a surface cleaner.

#### 3.6.1. Vaccines

Chicken keepers were aware of vaccines, generally understood to be a veterinary product given in water or injected before a chicken gets sick to prevent disease. The ability to consistently follow a recommended vaccination schedule depends on cost, packaging (vaccines sold in large quantities favor those with larger flocks), and administration. Participants in free-range production FGDs (10/12 FGDs) and semi-intensive production systems (12/12 FGDs) mentioned the use of vaccines as a preventive measure. The Newcastle disease (ND) vaccine was the most accessible due to its smaller package sizes, lower cost, and administration in drinking water. It was mentioned in 15 FGDs, with reports of vaccination in 13 FGDs; 85% of all participants reported access to the vaccine (Table 5). We did not ask individual participants about access to other vaccine types, choosing to interpret ND vaccine access as a proxy for access to preventive veterinary products in general. For example, we assumed the 15% unable to access ND vaccines would be unlikely to access other types of vaccines. 

The next two most used vaccines were Gumboro (reports of vaccination from six FGDs, five commercials, and one smallholder) and fowl pox (reports of vaccination from five FGDs, all commercial). Rarely mentioned vaccines were fowl typhoid (mentioned in two FGDs with reported use in one FGD), Marek’s disease (mentioned in two FGDs with no reported use), and infectious bronchitis (reported use in one FGD). Vaccines combining ND with infectious bronchitis are commercially available, so it is possible that the actual use is higher but was not explicitly mentioned by the participants. In the case of Marek’s disease, the prohibitive cost of the vaccine plus administration was reported as a barrier to use even for a hatchery. Vaccines that require an intramuscular or subcutaneous injection, such as the current fowl typhoid vaccine, or a web stab, such as fowl pox, are costly in part because of the need to engage a trained person to administer them.


*For most vaccines, I normally do it myself, but for injectables, I pay a vet who does it for me. That is for the wing stab and the intramuscular vaccine administered to the thigh. However, the ones we give in water, I give them myself. (Man, semi-intensive production, Kiambu County)*


Cold chain, particularly for vaccines, was less of a concern for chicken keepers in the urban and peri-urban study sites, with most participants traveling only a few kilometres to access veterinary products. Even in these environments, products requiring cold chain distribution reduce profit margins for agrovets because of the cost of storage and could be compromised by power outages. As a result, some chicken keepers, particularly those with semi-intensive production systems, “prefer and trust big institutions that can afford a backup system like a generator.” Chicken keepers in Machakos County faced greater problems than their neighbors near Nairobi, some traveling up to 50 km to access vaccines.

#### 3.6.2. Antibiotics

Antibiotics are easily administered in drinking water, require no refrigeration, and serve as a solution when the problem is uncertain. Antibiotics were mentioned in 17/24 FGDs and used by 69% of all participants within the previous three months. Chicken keepers described using antibiotics to both treat and prevent disease. No participants described using antibiotics as a growth promoter. Similarly, no participants reported using or being aware of feed products that contain antibiotics, although some did mention giving antibiotics to accompany transitions from one type of feed to another. “I look at the feces, and they are not good, I go for antibiotics” (Woman, semi-intensive production, Kiambu County). “When I see signs of sickness, I buy antibiotics. My project is small; I do not call a vet” (Man, free-range production, Nairobi County).

Although participants spoke about the purchase of antibiotics, few could recall the names of the antibiotics used. Antibiotics mentioned by name included Tylodoxin (for treatment of diarrhea mentioned by a man with semi-intensive production), Ecogin (an antibiotic and vitamin mix sold for growing chicks mentioned by a man with semi-intensive production), Sulfaclozine Sodium Monohydrate, or ESB 30% (for treatment of bloody diarrhea by a woman with semi-intensive production), Tylosin (for treatment of “snoring” or respiratory signs mentioned by a woman with free-range production), Biotrim (mentioned by women with free-range production “because it is cheap”), and Poltricin (mentioned by a man with free-range production). A complete list of products mentioned by FGD participants and key informants, including antibiotics, vaccines, traditional medicines, and cleaning products, can be found in Appendix A, Table A2.

The withdrawal period is defined as the period from when a drug, typically an antibiotic, is administered to the time when drug residues in eggs and meat have dropped below a specified threshold, allowing them to be safely consumed. Concerns about the withdrawal period were mentioned in all FGD types (11/24 FGDs). Chicken keepers often knew about the withdrawal period, worried about the human health effects of residues in meat and eggs, yet acknowledged they were unable to take on the burden of losses during the withdrawal period and often sold birds and eggs anyway. 


*I think our biggest challenge, especially when dealing with layers, is the withdrawal of eggs once the chickens have been medicated. Broilers are a bit better. You cannot expect me to collect my 35 trays of eggs for three days and dispose of them. I will sell the eggs. (Man, semi-intensive production, Nairobi County)*


Notably, concern about the withdrawal period and residues was not limited to antibiotics but was often mistakenly attributed to the use of a wider range of veterinary products, including vaccines, as shown in the conversation below. Vaccines, designed to stimulate the immune system’s response against disease, do not have withdrawal periods. 


*Participant: I went back to the vet and was told at least every month I should give the chickens medication, alternating between Newcastle and Gumboro vaccines.*



*Facilitator: Has this alternating and medicating monthly had any effect?*



*Participant: The medicine cost me and there was a withdrawal period for eggs and meat. (Woman, free-range production, Nairobi County)*


Participants across all four FGD types (13/24 FGDs) mentioned antibiotics or drugs not working. Possible explanations for this include using the incorrect drug, administration route, or dosing; expired or incorrectly stored products; and/or drug-resistant pathogens. Regardless of the reason, drugs not working contributed to additional expenses for the chicken keepers, distrust of animal health professionals or agrovet employees if they were engaged, and in some cases, a search for alternative solutions such as traditional medicine. A man with semi-intensive production in Nairobi County lamented the cost of drugs and expressed frustration with the level of training of the agrovet employee who sold him the product: “Drugs are available, but they are not effective; they have let us down. A drug may be expensive but very ineffective, and the person dispensing it also has no knowledge.”

Some participants used antibiotics against viral pathogens, which are ineffective. For example, four participants (two women and two men, both in a semi-intensive production system) described using antibiotics to treat Newcastle disease. “For the bigger chickens, I would say Newcastle disease affects them most. It is bad because when it strikes, it has no treatment. You may only use antibiotics to control the symptoms”, explained a man with semi-intensive production from Kiambu. A woman with semi-intensive production, also in Kiambu, described what happens when Newcastle disease enters her flock: “By the third day, I lose them. Even if I give them antibiotics, they still die.” A man in Nairobi County described his frustration with “agrovet drugs” in general, saying: “I use traditional remedies because agrovet drugs are not working.”

#### 3.6.3. Traditional Medicine

Use of traditional medicine to treat and prevent chicken diseases was reported by men and women in both production systems (21/24 FGDs). Traditional medicines were defined broadly to include plants such as aloe vera, foods such as ginger, and other substances such as wood ash. Traditional medicines were usually administered to chickens in drinking water, with a few applied topically for wounds, lesions, or external parasites. A man in Machakos County with semi-intensive production described the common scenario of relying on both veterinary products and traditional medicines as a chicken producer, saying: “We have the medicines we can use, such as antibiotics, and traditional ways to treat them, like wood ash and aloe vera.” Some participants used language and information borrowed from the veterinary profession; for example, a woman with semi-intensive production in Kiambu County said: “The Mexican marigold is an antibiotic. We also give garlic, which is a very strong antibiotic.”

Traditional medicines served as an affordable alternative to veterinary products and were appreciated for being safer for humans compared to veterinary products. “Aloe vera is planted at no cost. We also eat the chickens, and I do not want to eat one that has many drugs,” said a woman with free-range production in Kiambu County. A woman in Nairobi County, also with free-range production, went so far as to suggest using veterinary products will degrade the superior qualities of the local chicken if used, saying: “‘Yes, traditional medicine does not cost me, plus it is not pure *kienyeji* when I use the medication from an agrovet.”

The downsides of traditional medicine included uncertainty about dosing, challenges with administering to large flocks, and “not being completely effective.” See Table A2 in the Appendix A for a complete list of traditional medicines mentioned.

#### 3.6.4. Cleaning

In addition to veterinary applications, phage products could potentially be used as surface cleaners that target specific strains of bacteria. As described in Section 3.3., roles in routine and management tasks, cleaning, and hygiene were viewed as important parts of the disease prevention, and were mentioned more by female (six semi-intensive FGDs and five free-range FGDs) compared to male (four semi-intensive FGDs and two free-range FGDs) chicken keepers. Products used when disinfecting chicken houses included powdered washing soap, Norocleanse, glutraquat TH-4, and Jik. Disinfectants were mixed with water or sprayed using backpack sprayers designed for the application of pesticides and fungicides to crops. Less costly alternatives included wood ash, sprinkled in chicken housing to kill germs and insects, and soda ash mixed with water in a footbath at the entrance. 

### 3.7. Production Factors

In addition to veterinary considerations, chicken keepers mentioned four aspects of production that influence adoption opportunities for phages: regulatory environment, feed, housing, and biosecurity.

#### 3.7.1. Regulatory Environment

Government regulations surrounding the sale of veterinary products and feed, as well as the movement and disposal of animals, were mentioned by men and women with semi-intensive production and by men with free-range production. Participants in the men’s free-range production groups worried about fake products and hawkers repackaging feed and drugs under trusted brand names. They did not trust the Kenya Bureau of Standards to appropriately intervene. A man in Nairobi County explained:
*Now, we trust Unga feeds, but there are people who have used their packaging bags to sell substandard products, so I do not trust anything Kenyan. I may only trust it if it is from angels or something from a tree.*

Participants in semi-intensive production were concerned that internationally banned veterinary drugs may be sold in Kenya without consumers knowing and that vaccine quality would be adversely affected by poor handling.

#### 3.7.2. Feed

The high cost and unpredictable quality of packaged chicken feed were mentioned in almost every FGD. Women with free-range production reported purchasing feed as an area where their husbands often financially contributed. Participants in all FGDs complained that a company with high-quality feeds could start using poor-quality feeds (e.g., rotting maize), and that the packaging standards and regulatory processes could be weakened by corruption. They worried about aflatoxins and changes in formulations of supplements. They also described adverse health effects in their flocks, which they attributed to poor-quality feeds. A woman in Kiambu County with 700 layers said:
*Sometimes the feeds we give to the chicken cause the eggs to be small or the yolk to be light, and we are forced to buy the booster (a product containing antibiotics) because the feed is not the same. The company may be the same, but the feed is not the same.*

This contributed to many chicken keepers regularly changing feeds in an effort to maintain high production and reduce costs. Across all FGD types, people described formulating their own feed, often by purchasing ingredients from grain mills, with varying levels of success. 

#### 3.7.3. Housing

Well-constructed housing, with ventilation and sufficient space to avoid overcrowding, can prevent conditions conducive to bacterial and other infections. Unfortunately, as a livestock production officer in Machakos explained, some chicken keepers construct housing that contributes to diseases and possibly increases the use of antibiotics:
*Since they do not want them to be taken away by predators, they enclose them in a very small house. If any disease comes, there will be no ventilation, so that disease will kill the chicks. When they start treating, they will go and give just any antibiotic, so they can even overdose.*

Constraints to housing practices included the cost of building materials, the availability of land (especially for commercial keepers), knowledge about best designs, and, as mentioned above, the threat of predators, including mongooses and domestic dogs. Constructing housing was widely considered to be a man’s responsibility.

#### 3.7.4. Biosecurity

Women with free-range production reported more detailed biosecurity practices than men (4/6 FGDs), including cleaning the housing with disinfectants or soda ash, limiting visitors, constructing housing with permanent floors, and isolating sick chickens. Only one participant, located in Kiambu County, reported using a foot bath. Men with free-range production reported isolating sick birds and disinfecting housing through cleaning or spraying (3/6 FGDs). Some knew about foot baths, but few reported using them. Biosecurity practices were mentioned in all six of the women’s semi-intensive production FGDs, including spraying disinfectant at regular intervals, cleaning housing with disinfectants or soda ash, isolating sick chickens, using foot baths, minimizing visitors, and separating chicken housing by age group. Similar practices were mentioned by men with semi-intensive production (5/6 FGDs), with the addition of requiring changes of clothing and shoes from employees, assigning limited people to have access to certain buildings or areas, and concerns about veterinary professionals bringing disease from other farms.

### 3.8. Human Health

Since *Salmonella* can cause illness in humans, to assess awareness of zoonoses, participants were asked to describe any negative effects chicken keeping could have on human health. Across all FGDs, awareness of zoonoses was low (3/24 FGDs). Three women in two FGDs for free-range production mentioned people getting sick from chickens. One had a sick child and was told at the hospital it was caused by chickens; another mentioned touching chicken manure without gloves as a method of transmitting disease to people. A man keeping chickens commercially mentioned zoonoses but said it had not been a problem in his business. No one mentioned diarrhea, vomiting, food-borne illness, or any human health conditions by name. 

The most mentioned negative health concern was the effects of drugs and growth boosters in meat and eggs on human health (14/24 FGDs), followed by chickens attracting flies or parasites such as mites (11/24 FGDs), and asthma/allergies exacerbated by dust or chicken manure (10/24 FGDs). The withdrawal period was mentioned by participants in all four FGD types and in 11/24 FGDs. Nairobi County participants mentioned the withdrawal period more than participants from other counties (6/8 FGDs). For example, withdrawal was mentioned by only two FGDs in Machakos, both commercial producers. Most participants did not specify or know what types of veterinary products have withdrawal periods, with some participants incorrectly associating withdrawal periods with vaccine use or veterinary inputs in general.


*Now that I keep kienyeji chickens, I mostly do not want to use medication because they are eaten by the family, and when someone comes to buy them, they ask many questions, specifically if the chicken has been given any medication. We want to sell the products that we also eat. That is why I give them aloe vera, because even humans use aloe vera. The times I use antibiotics, it is not because I want to, but because there is no other way or other expertise I can use. (Woman, free-range production, Nairobi County)*


## 4. Discussion

Overall, we sought to explore what options exist for designing bacteriophage products that can be adopted by female and male chicken farmers in both free-range and semi-intensive production systems. The following design recommendations consider both gender and production systems.

A social lean canvas summary describes how a phage product can be a solution to stated problems and its unique value propositions for female and male chicken keepers in free-range production systems (Figure 1). 

With lower budgets for veterinary inputs, less access to veterinarian services, and low biosecurity in free-range production, participants described seeking treatments rather than preventive actions, such as adhering to vaccination schedules. In this context, a phage product that can treat fowl typhoid would be valued. The Newcastle disease vaccine was found to be the only preventive veterinary input often used by both women and men respondents across the two systems and by more women (89%) than men (65%) in free-range systems. We therefore recommend a second product, marketed as a vaccine, that combines phages with ND vaccines for disease prevention. Most veterinary inputs, from antibiotics to ND vaccines, are administered through drinking water rather than intramuscular injections, so it is recommended that all phage-based products, whether marketed as vaccines or treatments, be administered orally. This allows the chicken keepers to depend less on costly administration by veterinarians. Alternatives to treatment and prevention of bacterial diseases included cleaning and biosecurity practices, most mentioned by women, and the construction of appropriate housing, largely the responsibility of men.

Women in free-range systems described having less access to capital for inputs such as feed and veterinary services/products and often depending on their husbands to purchase veterinary products, both because of cost and their greater access to agrovets and veterinarians due to more responsibilities outside the home. Even though men may bear the cost of the inputs, women took an active role in overseeing or initiating the purchase of those inputs. Notably, women in free-range systems described disease problems using more descriptive signs than disease names, such as white diarrhea (associated with the pullorum disease) or reduced laying (associated with fowl typhoid, fowl coryza, and *E. coli*). It may be beneficial to consider including disease signs on packaging or marketing materials. Participants preferred small package sizes appropriate for those with smaller flocks and clear dosing instructions, with dosage cups or marked packaging to indicate quantities. Lastly, given concern about residues from veterinary products in meat and eggs, a phage product should be labeled and marketed as having no withdrawal period and no residues passing into eggs or meat. 

Both women and men preferred rearing local and improved local breeds of chickens because they perceived them to be more disease-resistant and requiring minimal veterinary inputs. Such a perception could limit the adoption of any veterinary product without the sensitization that local chickens can benefit from veterinary interventions. Local chickens are not only kept by free-range producers or in rural areas, but also by semi-intensive and some emergent commercial farmers since there is a demand for meat and eggs from local chickens at the Kenyan market. 

Women and men in semi-intensive production systems reported using more types of vaccines, such as Gumboro and fowl pox; however, fowl typhoid vaccines were among the least accessible because of the cost and technical skill needed to administer an intramuscular injection. In this context, in addition to the design suggestions for free-range chicken keepers, a phage product marketed as a vaccine to prevent fowl typhoid would be valued (Figure 2).

In semi-intensive production systems, in comparison to women, men delegated routine tasks mostly to their wives and employees. Furthermore, women and older men tended to perform the work themselves or with the assistance of children. To prevent the burden of additional labor administering medication or vaccines, which may fall disproportionately on women, oral delivery in drinking water is the most practical delivery method for any product. Concerns about feed quality led to many participants changing feed brands frequently and aspiring to formulate their own feeds; therefore, delivery of phages through feed is not recommended for the Kenyan context. Hatcheries and multipliers, informal businesses that raise hatchery chicks into teen chicks for sale to livestock keepers, could be targeted as early adopters of a phage product due to their large numbers of chickens and the risk of bacterial diseases due to overcrowding.

Women in semi-intensive systems were the most likely to keep layers, saying the marketing of eggs was compatible with their household duties compared to selling broilers. This makes them most financially impacted by reduced laying, which is associated with some bacterial diseases, with potentially more to gain from control of disease through a phage product. Men in semi-intensive systems mentioned parasites and coccidiosis much more than other participant types, suggesting that in a resource-constrained environment, they may have other priorities that supersede control of bacterial diseases. Urban consumers of meat and eggs in Nairobi preferred animal products from local chickens in part due to concerns about drug residues, so the lack of a withdrawal period for phages continues to be a unique value proposition compared to antibiotics. In contrast, awareness of zoonoses was low, so presenting phages as a way to reduce human health problems such as diarrhea may not gain traction. 

One potential application of phages considered in the study was their use in cleaning products. Existing cleaning products, including soap and bleach, are accessible and affordable, and many kill a variety of pathogens. Therefore phage products as surface cleaners, while being eco-friendly, will probably not give any additional advantages to chicken keepers in free-range and semi-intensive systems. Phage-based cleaning products may be more useful in a more commercial context; however, that is beyond the scope of this research. Hatcheries and multipliers, informal businesses that raise hatchery chicks into teen chicks for sale to livestock keepers, could be targeted as early adopters of a phage product due to their large numbers of chickens and the risk of bacterial diseases due to overcrowding. 

Lastly, financial incentives for agrovet employees are a consideration for the uptake of a new veterinary product. Agrovets were the most mentioned source of veterinary care for all participant types except women in semi-intensive production, who mentioned veterinarians more. This is consistent with the findings of a study in five Sub-Saharan African countries showing 80% of livestock-keeping households acquired antibiotics from agrovets [16]. Considerations for agrovet employees when stocking products include profit margins, shelf life, and cold chain requirements. The closer a phage product can be to an affordable antibiotic with a long shelf life and no need for refrigeration, the more competitive it will be. 

Phage therapy is being investigated to treat *Salmonella* infections in chickens globally. The co-creation process between the laboratory research team and the socioeconomic/gender team is an example of best practices in product development, increasing the likelihood the final product will be adopted and useful to a diversity of chicken keepers. Strengths of this study include early collaboration between the product development and social science teams and providing open access to qualitative data. Limitations of the study include the inability to look at adoption opportunities for more commercialized chicken keepers due to the focus on sector 3 and 4 participants with moderate flock sizes and the inability to consider intra-household or spousal dynamics for veterinary decision-making due to the structure of the FGDs with one individual representing each household. The focus on a limited geographical area and greater depth with fewer participants limit the ability to generalize the needs and priorities of chicken keepers in other contexts. The focus on peri-urban Nairobi and the selection of women participants by community mobilizers, who may over-represent successful female chicken keepers, may suggest more egalitarian attitudes than would be found in more rural parts of Kenya. 

## 5. Conclusions

Next steps for a phage product in Kenya include ongoing research, such as biological characterization of known or novel phages towards Kenyan *Salmonella* strains, testing efficacy and safety, exploring therapeutic effects of phages in combination with existing products, such as antibiotics and Newcastle vaccines, and continuing conversations about the registration process with the Veterinary Medicines Board. Some of the language in this article, such as the suggestion “to market a product as a vaccine” (i.e., implying that the vaccine should be administered before chickens have clinical signs), may be in conflict with registration requirements or definitions. We recommend marketing and labeling strategies that help veterinarians, agrovet employees, and chicken keepers understand how the new product will be used. It may also be appropriate to register multiple products, such as one used for treatment and another used prophylactically, as a way to best serve the different needs of chicken keepers in Kenya. 

## Figures and Tables

**Figure 1 viruses-15-00746-f001:**
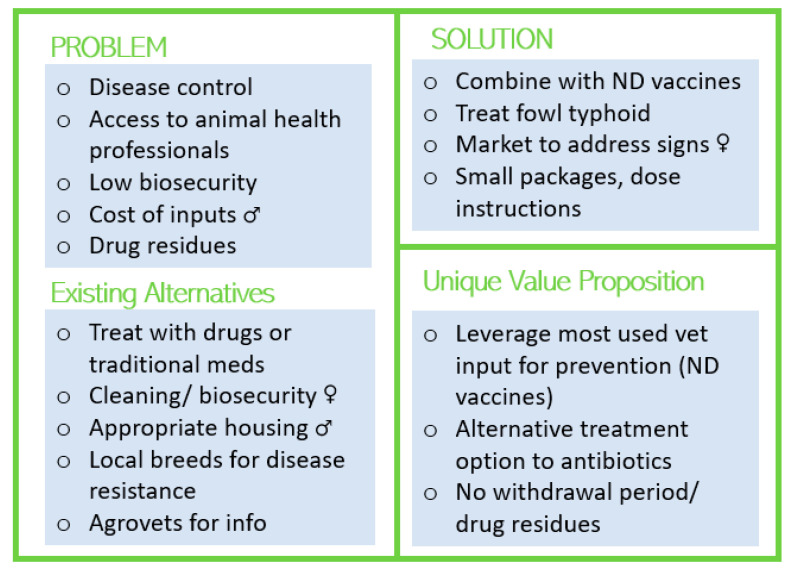
Lean social canvas summary for chicken keepers with free-range production systems (<50 chickens). Symbols indicate points that differentially influence women (♀) or men (♂).

**Figure 2 viruses-15-00746-f002:**
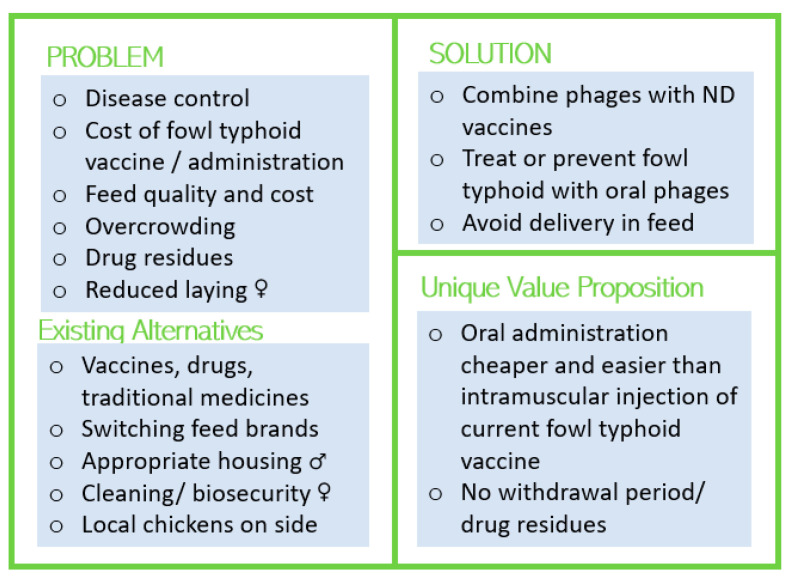
Lean social canvas summary for chicken keepers with semi-intensive production systems (>50 chickens). Symbols indicate points that differentially influence women (♀) or men (♂).

**Table 1 viruses-15-00746-t001:** Number of participants across the 24 focus group discussions broken down by site, gender, and production system.

County	Sub-County	Free-Range ^1^		Semi-Intensive		Total
		Women	Men	Women	Men	
Nairobi	Ruai	7	5	7	8	27
	Dagoreti	10	7	6	8	31
Kiambu	Thika Makongeni	6	5	8	4	23
	Gatundu South	6	7	7	5	25
Machakos	Machakos Central	9	7	8	7	31
	Mwala	6	6	5	8	25
Total		44	37	41	40	162

^1^ Free-range defined as <50 chickens and semi-intensive as >50 chickens.

**Table 2 viruses-15-00746-t002:** Focus group participant characteristics by production system and gender.

		Free-Range ^1^		Semi-Intensive		Total
		Women	Men	Women	Men	
Education	Primary	12	6	5	4	27
	Secondary	23	19	23	18	83
	Above	9	12	13	18	52
Age (yrs) ^2^	<35	6	9	4	10	29
	35–60	36	20	29	24	109
	>60	2	8	7	6	23
Marital status	Married	38	29	35	38	140
	Single	4	8	3	1	16
	Widowed	2	0	3	1	6

^1^ Free-range defined as <50 chickens and semi-intensive as >50 chickens. ^2^ One participant declined to provide their age. n = 162 participants.

**Table 3 viruses-15-00746-t003:** Flock size and type of chickens of focus group participants.

		Free-Range ^1^		Semi-Intensive		Total
		Women	Men	Women	Men	
Flock size	Mean	44	30	397	247	180
	Median	39	26	250	70	55
Type of chicken ^2^	Local	30	21	8	8	67
	Improved	5	7	8	5	25
	Layers	0	0	6	2	8
	Broilers	0	0	4	1	5
	Multiple	9	5	15	24	53

^1^ Free-range defined as <50 chickens and semi-intensive as > 50 chickens. ^2^ Four participants did not provide information about the type of chicken kept. n = 162 participants.

**Table 4 viruses-15-00746-t004:** Number of focus group discussions (FGDs) mentioning diseases, syndromes, and signs.

	Free-Range ^1^		Semi-Intensive		Total
	Women	Men	Women	Men	
Newcastle disease	4	2	5	5	16
Respiratory signs	4	2	5	4	15
Coccidiosis ^2^	4	2	2	5	13
Parasites	4	1	3	5	13
Fowl typhoid	1	2	2	1	6
Salmonellosis	1	1	1	0	3
Pullorum disease	0	0	0	0	0
White diarrhea	5	2	3	1	11
Chick mortality	2	2	2	1	7
Reduced laying	2	0	1	1	4

^1^ Free-range defined as <50 chickens and semi-intensive as >50 chickens. ^2^ Bloody or red diarrhea was coded as coccidiosis. Each category had six FGDs for a total of 24 FGDs. Highlighted rows indicate bacterial diseases and associated signs.

**Table 5 viruses-15-00746-t005:** Information about veterinary care and services as reported by focus group participants.

		Free-Range ^1^		Semi-Intensive		Total
		Women	Men	Women	Men	
Source vet care ^2^	Agrovet	24	19	17	20	80
	Vet	13	12	23	15	63
	Community	2	3	1	4	10
	None	2	2	0	1	5
Access to ND vaccines (%)		89%	65%	100%	83%	85%
Used antibiotics last 4 months		73%	59%	68%	73%	69%

^1^ Free-range defined as <50 chickens and semi-intensive as >50 chickens^. 2^ Four responses for the other. n = 162 participants.

## Data Availability

Supporting data can be downloaded at: https://data.mel.cgiar.org/dataset.xhtml?persistentId=hdl:20.500.11766.1/FK2/QKORYG (accessed on 9 March 2023). Dataset: Focus group discussion transcripts 1–24.

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
