# Peer review of "Gender-Responsive Design of Bacteriophage Products to Enhance Adoption by Chicken Keepers in Kenya"

_viruses, 2023, doi:10.3390/v15030746_

Round 1

Reviewer 1 Report

This is a very well written article based on a soundly designed study. I thoroughly enjoyed the way the authors use qualitative data and analysis to support a rigorous research and co-design process and produce sound recommendations. I have three comments (and minor editorial suggestions), which I would like the authors to take into consideration in order to improve the manuscript. 

First, I believe the title does not convey the full value of the paper (I was myself about to refuse the invitation to review it!). I would suggest “Gender-sensitive design for bacteriophage products…” or “Design opportunities… by women and men Kenyan chicken keepers” or similar. My point is that “gender” or “women” terms in the title do justice to the gender approach that informed the analysis (which is fundamental even if the authors state that a gender-specific design is not warranted in the end).

Second, I would suggest the incorporation of gender-specific differences into recommendations for product design. While I understand that differentiations by production system were found to be more important in peri-urban Kenya, I think the choice not to incorporate gender differences (lines 843-46; 912-14) is unduly restrictive. Maybe a 2x2 table that considers the interplay between two production systems and two genders would be feasible and desirable? Without being familiar with these products, I dare propose two examples. Since women seem to use surface cleaners more frequently than men (see 720-22), this option can be perhaps more attractive to women, especially in the semi-intensive production system. If so, then the statement in lines 875-9 could be changed (it also seems contradictory to earlier points). Also, given lower access to capital, family labor and veterinary care by women in free-range production system, one may suggest that out of the two options suggested in line 848 - phages with ND vaccines for disease prevention and/ or use phages to treat fowl typhoid and other bacterial diseases – the second would be more attractive to women in this group. And so forth. 

Third, to add to the discussion section (which raises important points regarding labeling of products and promotional materials 886-895), the authors could add a comment or a sentence speaking to the type of information promotion that could support a CORRECT use of this new product - given the finding that chicken keepers lack trust in agrovets and lack specific knowledge about medicines.  

Minor editorial suggestions follow:

Line 159: Add (see breakdown below) to this sentence to indicate more information will be provided about the 17 key informants.

190: Specify what "these" refer to: FGD?

Author Response

Please see our response in the attached pdf.

Reviewer 2 Report

Major & reviewer questions:

1.      What is the problem of antibiotic resistance of Salmonella isolated from poultry in Kenya? What percentage of Salmonella isolates are MDR? What antibiotics are most likely to be resistant to? It should be shown that phages should be an alternative to antibiotics, especially broad-spectrum ones. The authors should make it clear how much of a problem its widespread antibiotic resistance is today.

2.      Can phage therapy be used in veterinary or animal husbandry in Kenya? (legal conditions)

3.      If yes, what phage preparations are used / registered / available for Salmonella control in poultry farms? If not, the article still lacks a chapter on available phage preparations aimed at eradication of Salmonella strains.

4.      In my opinion, the title of this article does not correspond to the content of the manuscript. I have the impression that phage preparations are just an addition, which the Authors mention in the Introduction and only later in the Discussion Section. This aspect of the possibility of using phage therapy should be more exposed.

5.      In the research questions, I did not notice those about phages. I have not found the opinion of poultry farmers regarding the potential inclusion of phage biopreparations in veterinary products.

6.      I believe that this article may be very interesting and necessary, but it needs rewriting.

Minor:

1.      Keywords: individual keywords should be separated by a semicolon; no dot at the end

2.      In the manuscript, the form of citations should be corrected, in accordance with the recommendations of the Journal (e.g. in lines 50, 67, 71, …)

3.      In line 109 - should be [… (p < 0.05) …]

4.      References should be corrected in accordance with the recommendations of the Journal. Furthermore, the manuscript contains 34 pages and only 37 references; 32% of which were published before 2014.

Round 2

Reviewer 2 Report

The Authors have adequately responded to my queries. I have no further comments.